# Constraining Dark Boson Decay Using Neutron Stars

**Wasif Husain, Dipan Sengupta * and A. W. Thomas**

ARC Centre of Excellence for Dark Matter Particle Physics, The University of Adelaide,
Adelaide, SA 5005, Australia; wasif.husain@adelade.edu.au (W.H.); anthony.thomas@adelaide.edu.au (A.W.T.)
* Correspondence: dipan.sengupta@adelaide.edu.au

**Abstract:** Inspired by the well-known anomaly in the lifetime of the neutron, we investigated its consequences inside neutron stars. We first assessed the viability of the neutron decay hypothesis suggested by Fornal and Grinstein within neutron stars, in terms of the equation of state and compatibility with observed properties. This was followed by an investigation of the constraint information on neutron star cooling that can be placed on the decay rate of the dark boson into standard model particles, in the context of various BSM ideas.

**Keywords:** neutron star; neutron decay; dark matter; dark boson decay; dark fermions

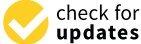



## 1. Introduction

Neutrons are a fundamental constituent of our universe. It has been over almost a century since they were discovered, but their lifetime [1] still presents a challenging problem to solve. In particular, current experiments appear to show a difference in the neutron lifetime when measured with different methods. In the bottle method [2–5], the neutrons are trapped and the number counted after a fixed time, with no specific determination of the decay mode. In contrast, using the beam method [6–8], one actually observes the protons produced in $\beta$ decay. Of course, the lifetime of the neutron should be the same, regardless of the method of measurement. However, the lifetime of the neutron shows a discrepancy. Using the bottle method, Reference [9] found the lifetime of the neutron to be $877.75 \pm 0.28_{stat} + 0.22 - 0.16_{syst}$ s, which is very close to the lifetime measured in [2,3]. On the other hand, using the beam method, the lifetime has been measured to be $887.7 \pm 0.7\,(stat) + 0.4/-0.2\,(sys)$ s [3].

A resolution of this discrepancy in the lifetime of neutrons could potentially lead to new physics. A solution along those lines was recently proposed by Fornal and Grinstein [10–12], who proposed an extra decay channel of the neutron into dark matter. Based on the difference in lifetimes, they suggested that roughly 1% of the time, neutrons decay into dark matter, while for the remaining 99% of the time, they undergo $\beta$ decay. In the beam method, the dark matter would go undetected and uncounted, while in the bottle method, the effect of dark matter is automatically included.

According to the hypothesis of Fornal and Grinstein, the dark decay mode of the neutron is

$$n \longrightarrow \chi + \phi \,, \tag{1}$$

where $\chi$ is the dark fermion and $\phi$ the dark boson. This hypothesis has attracted the interest of many physicists. For example, experimental studies showed very quickly that the $\phi$ particle could not be a photon [13,14]. The hypothesis was also very rapidly subjected to tests using the properties neutron stars, which indicated that dark fermions, $\chi$, have to experience a strong vector repulsion in order to be consistent with the observations [15–18]. A recent study [19] suggested that there might be an observable signal of this decay if one could observe the neutron star right after its birth. An interesting discussion about the neutron decay can be found in [20].

An alternative to the Fornal and Grinstein hypothesis was proposed by Strumia in [21], where the author suggested that neutrons might decay into three identical dark fermions, $\chi$:

$$n \longrightarrow \chi + \chi + \chi, \tag{2}$$

with each of them having baryon number 1/3 and mass $m_\chi$ = (mass of neutron)/3. An advantage of this proposal was that, to be consistent with the constraint on the maximum mass of neutron stars, dark fermions, $\chi$, are not required to be self-interacting. Our recent study in [22] on the Strumia hypothesis agrees with this claim and indicated that one could find observable signals of neutron decay similar to those found within the Fornal and Grinstein hypothesis.

Within the Fornal and Grinstein hypothesis, shown in Equation (1), the mass of the decay products must be in the range 937.9 MeV < $m_\chi + m_\phi$ < 938.7 MeV for the known stable nuclei to remain stable [10,11]. To date, most of the studies on the hypothesis using neutron stars have considered the $\phi$ as an extremely light particle, which escapes the neutron star immediately, and treating $\chi$ as almost degenerate with the neutron. There is a possibility that, if the $\phi$ boson has a mass close to the difference of the masses of the neutron and $\chi$, for example of order 1 MeV, then it may remain trapped inside the neutron star.

Here, the focus was on trapped $\phi$ bosons and their effects on neutron star heating. This led to a strong constraint on the lifetime of $\phi$, which was then compared with the limits from other studies of dark matter candidates. The manuscript is divided into sections as follows. Section 2 covers the necessary model for the equation of state of nuclear matter inside the neutron star and explains the change associated with neutron decay into $\chi$ and $\phi$. This is followed by Section 3, where the consequences of trapping the $\phi$ boson are explored. In Section 4, the decay modes of the $\phi$ boson are studied in detail. Finally, Section 5 presents a summary of our findings.

**2. Neutron Stars**

Neutron stars are comparatively small objects that come into existence when an ordinary star of mass 8–15 M$_\odot$ dies. Neutrons are unsurprisingly the dominant component of a neutron star, and if neutrons decay into $\chi$ and $\phi$, then this decay must also take place inside the neutron star. Therefore, neutron stars must contain $\chi$ fermions and, in the circumstances explained earlier, also $\phi$ bosons, and their presence inside neutron stars must change their properties [18,23–41]. There are some strong constraints on the properties of neutron stars imposed by observations that a realistic neutron star model must follow. For example, Reference [42] showed that a neutron star of mass 1.4 M$_\odot$ should have a radius 10–14 km. PSR J1614-2230 [43] and PSR J0348 + 0432 [44] have masses of 1.928 and 2.01 M$_\odot$, respectively, so the neutron star model must predict the maximum mass of neutron stars to be at least 2 M$_\odot$ [1]. The tidal deformability should be consistent with the discovery [49,50] of gravitational wave detection by the LIGO and VIRGO observatories. Therefore, neutrons stars can be very helpful in testing the Fornal and Grinstein hypothesis.

Neutron star interiors cover a wide range of densities right from the surface to the core [51–84], with the cores containing the most-dense matter in the universe. To model the neutron star, one needs to adopt a suitable equation of state for the nuclear matter. At the core of a neutron star, the density could be as high as six-times the density of normal matter. Therefore, one needs to chose a model capable of describing the physics at such high densities. In this study, the quark meson coupling (QMC) model [85–88] was adopted to model the neutron star matter. A brief description of the QMC model is presented below.

*2.1. Quark Meson Coupling Model*

The quark meson coupling model was initially proposed by Guichon [85] and further developed by Guichon, Thomas, and collaborators [86,89]. In this model, the nucleons are treated as a collection of three quarks confined in an MIT bag [90]. The internal structure of the nucleon is treated with great importance, unlike other models, where nucleons are considered as point-like objects. In the QMC model, the interaction between baryons is

generated by the exchange of mesons, which couple self-consistently with the confined quarks. The strong scalar mean field in particular drives significant changes in the structure of the bound baryons.

The equation of state based on the QMC model has been shown to lead to an acceptable description of neutron star properties [22,84,88]. The effective mass of the nucleon in-medium may be expressed in terms of the scalar polarisability, "$d$", the mass of the free nucleon, $M_N$, and the coupling constant of the $\sigma$ field to the nucleon in free space, $g_\sigma$, as

$$M_N^*(\sigma) = M_N - g_\sigma \sigma + \frac{d(g_\sigma \sigma)^2}{2}. \tag{3}$$

The details of the QMC model can be found in [85,86,91]. For simplicity, in this study, it was assumed that neutron stars do not contain hyperons or strange matter at the higher energy densities; a nucleon-only equation of state is used [19].

### 2.2. Formalism including Neutron Decay

According to the hypothesis given in Equation (1), neutron stars must contain $\chi$ and $\phi$. As mentioned above, there is no a priori constraint on the mass of $\phi$, within a small window allowed. We chose to study the case $m_\phi$ = 1 MeV and $m_\chi$ = 937.7 MeV in this work, where $m_\phi$ is the mass of the $\phi$ boson and $m_\chi$ the mass of the $\chi$ fermion. In this case, the velocity of the $\phi$ boson is sufficiently low that it will be trapped inside the neutron star.

Inside the neutron star, $\phi$ bosons must condense, according to the Bose–Einstein condensation theory [19,92]. However, the total contribution of $\phi$ will be far too small to make any significant changes in the existing mass, radius, and tidal deformability constraints, even after condensation. In fact, we will see in later sections that the contribution of the $\phi$ bosons to the total mass is only about $1/10^6$ M$_\odot$. Although this is small, nevertheless, the amount of mass may contribute to the heating of the neutron star if the $\phi$ bosons decay into standard model particles. Therefore, the focus of this study was on $\phi$ boson decay into standard model particles inside neutron stars.

The equations were solved using the Hartree–Fock approximation. The full Hartree–Fock terms can be found in [19,92,93]. Although the $\phi$ bosons will be in the lowest possible quantum state, the dark fermions will constitute a gas of fermions. The dark fermions were assumed to be self-interacting, in order to survive against the observational constraints on the neutron star's properties. The self-interaction of the dark fermions was assumed to be similar to the neutron–$\omega$ interaction.

The presence of $\chi$ and $\phi$ inside neutron stars changes their composition. Therefore, the chemical equilibrium equations are [15,16]

$$\mu_n = \mu_\chi + m_\phi \qquad \mu_n = \mu_p + \mu_e \qquad \mu_\mu = \mu_e \qquad n_p = n_e + n_\mu \tag{4}$$

where $\mu$ represents the chemical potential of the different associated particles and $n_p$, $n_p$, and $n_e$ stand for the number of neutrons, protons, and electrons. The dark fermions, dark bosons, and nuclear matter particles were assumed to not interact with each other. Therefore, the neutron star contains two non-interacting fluids. However, because the contribution of dark matter compared to nuclear matter is very small, using the two-fluid TOV equation is not necessary. Therefore, for the ease of calculation, a one-fluid TOV was used.

### 2.3. Tolman–Oppenheimer–Volkoff Equations

To calculate the properties of the neutron star, the equation of state was combined with the structural equations derived using Einstein's equations of general relativity. Therefore, the TOV equations [26,27,94,95], given as

$$\frac{dP}{dr} = -\frac{[\epsilon(r) + P(r)][4\pi r^3(P(r) + P(r)) + m(r)]}{r^2(1 - \frac{2m(r)}{r})}, \tag{5}$$

$$m(r) = 4\pi \int_0^r dr.r^2\epsilon(r), \tag{6}$$

were integrated from the centre of the neutron star towards the surface, using the boundary conditions that, at the surface, the pressure and energy density should be zero. Here, $P$ is the total pressure, $P = P_{nucl} + P_{DM}$, and $\epsilon = \epsilon_{nucl} + \epsilon_{DM}$ is the total energy density, including the energy density of nuclear matter and dark matter. The tidal deformability was calculated by using the method explained in [96,97].

## 3. Results

In this section the consequences of the neutron decay on the properties of the neutron stars are given. The vector interaction of the dark fermions was increased until it followed the constraint [11,16,17,21,98–100] on the properties of the neutron stars.

As shown in Figure 1, the mass of the neutron star was reduced after the neutron decay. In fact, the maximum mass of the neutron star fell below 2 $M_\odot$ after the neutron decay [15,17] if the dark fermions were considered to be non-self-interacting. However, neutron stars of mass above 2 $M_\odot$ [43,44,101] have been observed. Therefore, in order to survive, a neutron star model must predict neutron stars of a maximum mass of at least 2 $M_\odot$. Figure 1 indicates that the dark fermions must have self-repulsion with a strength parameter of order 26 fm$^2$ to be consistent with the observations. Moreover, there is a significant reduction in the radius of the neutron stars after the decay, which suggests that neutron stars should spin up during the decay.

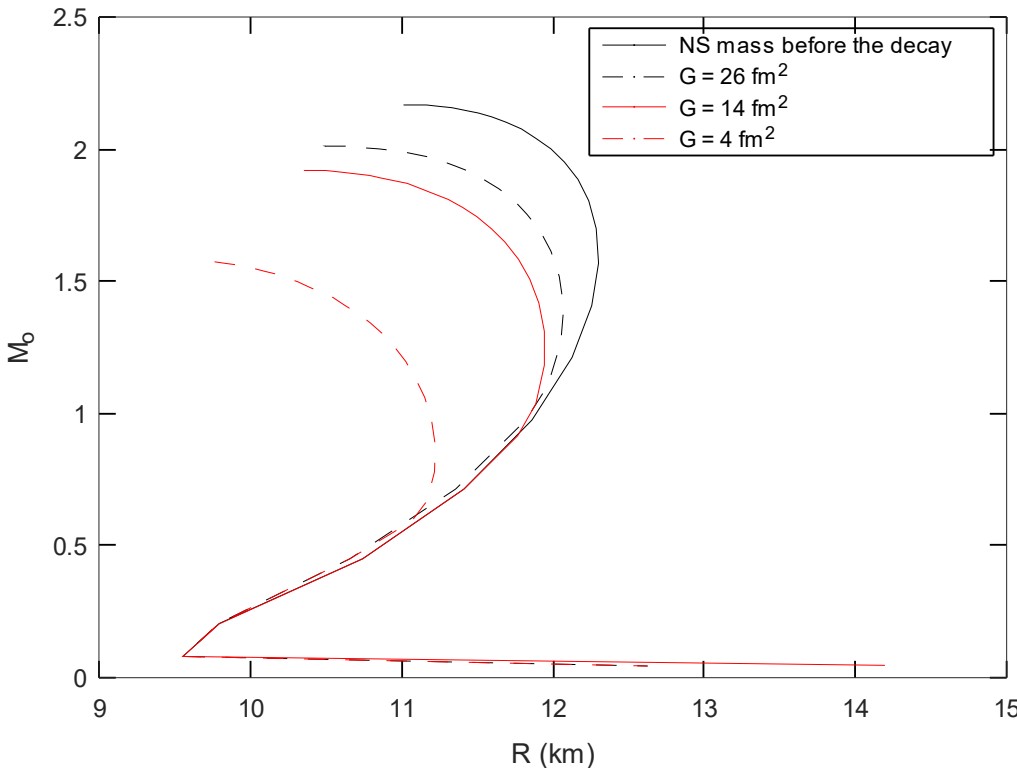

**Figure 1.** The total mass (given in solar masses) vs. the radius of the neutron star. Here, $G$ is the $\chi$-$\chi$ vector repulsion strength, which is increased to satisfy the observational constraints.

Figure 2 shows the tidal deformability against the radius of the neutron star. The analysis of the gravitational waves [49,50,102] indicated that a neutron star of mass 1.4 $M_\odot$ must have tidal deformability in the range 70–580, with a 90% confidence level. Figure 2 shows that dark fermions with a vector self-repulsion of strength 26 fm$^2$ satisfy the constraint on mass and tidal deformability.

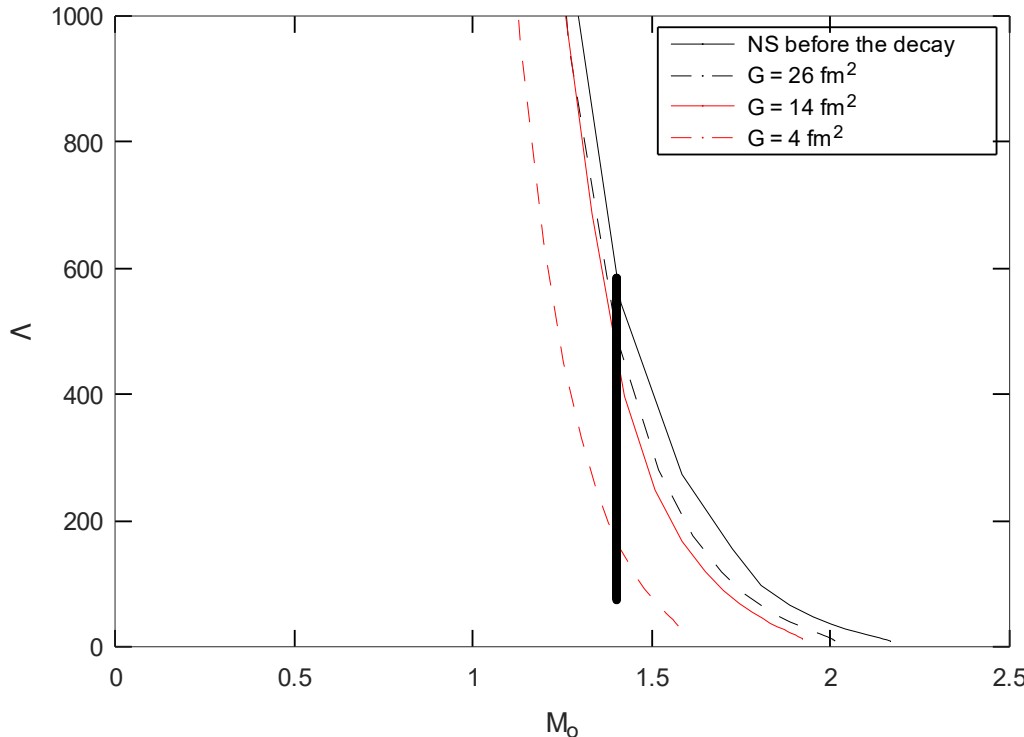

**Figure 2.** Tidal deformability against the mass of the neutron star. Here, *G* is the vector repulsion strength of dark fermions. The bold, black, vertical line indicates the acceptable range of values for tidal deformability [46,48].

Figure 3 shows the moment of inertia against the mass of the neutron star. The moment of inertia reduced after the neutron decay. When the dark fermion vector interaction was lowered, the difference in the moment of inertia increased. Thus, we were only interested in the case when the dark fermions, $\chi$, had a vector interaction strength $\geq 26$ fm$^2$. Figure 3 indicates that the moment of inertia of heavier neutron stars significantly reduced even when $G \geq 26$ fm$^2$, which should result in the spinning up of the neutron star. That, in turn, may provide a signal of the neutron's exotic decay.

Most studies indicate that neutron stars cool down very quickly by the standard Urca process. After approximately a million years, the neutron stars have a luminosity of order $10^{31.5}$ erg/s. Therefore, if the $\phi$ boson decays into photons, it will contribute to the heating of the neutron star, and after a million years, it must not contribute a luminosity $\geq 10^{31.5}$ erg/s. Based on the luminosity after one million years, we found that the lifetime ($\tau$) of the $\phi$ bosons must be greater than $1.85 \times 10^{11}$ years. With such a long lifetime, the luminosity stays essentially constant.

In the next section, the consequences of $\phi$ bosons decaying into standard model particles are explored.

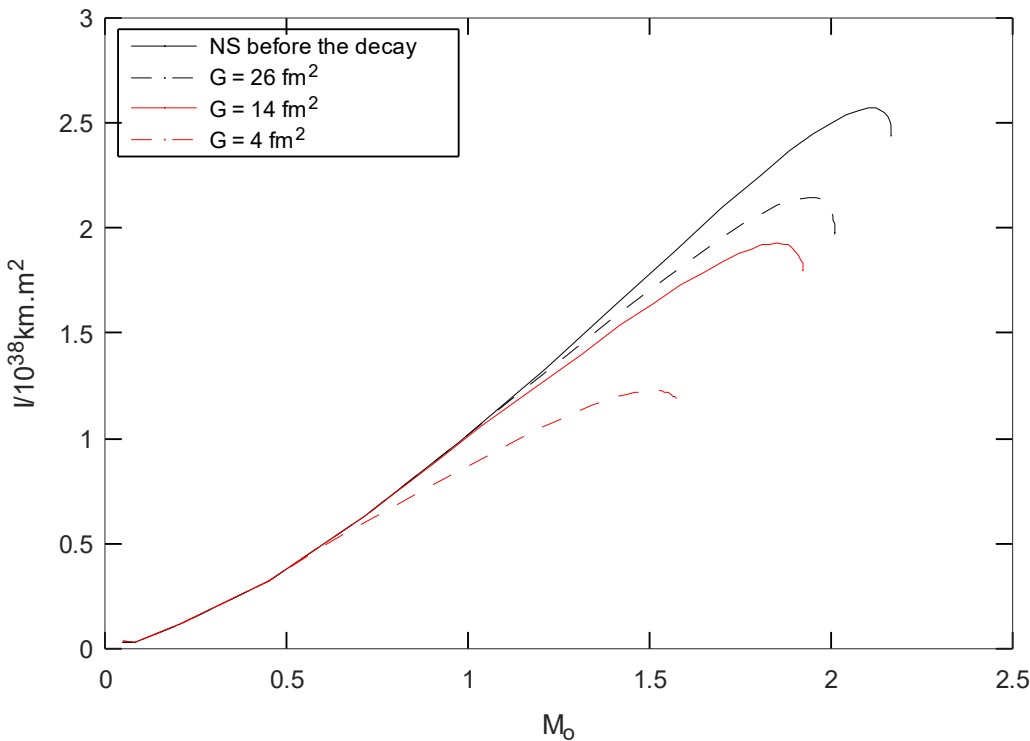

**Figure 3.** Total mass versus the moment of inertia of the neutron star with different self-interaction strengths of $\chi$s.

## 4. Decay Modes of $\phi$ Bosons

The decay products of the neutron, $\chi$ and $\phi$, are beyond-the-Standard-Model particles (BSM), which can originate from some UV-complete theory or be considered within some low-energy effective theory. While remaining agnostic about their origin, we can comment on the constraints on them from a variety of sources. The massive fermion, $\chi$, is an ideal candidate for dark matter and can form the bulk or all of the observed relic density today. We leave a detailed discussion on the details of this mechanism for a later expanded work. The other product of the decay, the boson, can originate from a BSM source. On general grounds and experimental considerations, the possibility that the boson is a photon has been ruled out. Here, we considered some simple possibilities for the bosons to couple to SM particles and constraints on the basis of the findings in the previous sections.

### 4.1. Scalars and Pseudoscalars

In the last few years, light scalar and pseudo-scalar particles have emerged as leading new physics candidates, which can be constrained from a variety of sources. While the primary motivation is derived from axions, simplified models with light scalars or pseudo-scalars have triggered much attention. Here, we assess their viability given our findings above.

The first bosonic candidate is a scalar coupled to the electromagnetic field strength:

$$\mathcal{L}_{int} = \frac{C_s}{\Lambda}\phi F_{\mu\nu}F^{\mu\nu} + \frac{m_f}{\Lambda}\phi\bar{f}f + \cdots \tag{7}$$

where $\phi$ is the scalar field, $F_{\mu\nu}$ the electromagnetic field strength, and $f$ the Dirac spinor for the leptons. The overall normalisation $\frac{C_s}{\Lambda}$ is model-dependent, while $m_l$ is the mass of the

lepton. The linear couplings can be generated by the scalar coupling to Higgs, as $\phi H^\dagger H$. A quadratic coupling can also be generated if $\phi$ carries a $Z_2$ symmetry:

$$\mathcal{L} = \frac{C_q}{\Lambda_q^2}\phi^2 F_{\mu\nu}F^{\mu\nu} + \sum_f \frac{m_f}{\Lambda_q^2}\phi^2 \bar{f}f + \cdots \tag{8}$$

In both cases, the dots indicate any other couplings that may be induced.

Couplings to the neutron can be obtained by integrating out, for example, heavy fermions, yielding dimension six operators, such that the effective neutron coupling can be written as:

$$\mathcal{L} \in L_{kin} + \lambda_{eff}n\chi\phi. \tag{9}$$

The linear (and quadratic) couplings induce a shift in the electromagnetic couplings that can be constrained from a variety of sources. A summary of these can be found in [103].

The next possibility is that of a pseudoscalar that couples, like an axion (like particle), to photons and derivatively to electrons:

$$\mathcal{L}_{int} = \frac{C_{s\gamma}}{\Lambda}\phi F_{\mu\nu}\tilde{F}^{\mu\nu} + \frac{C_f}{2\Lambda}(\partial_\mu\phi)\bar{f}\gamma^\mu\gamma^5 f + \cdots \tag{10}$$

For axion-like particles, in Equation (10), the effective ALP coupling to leptons generates a coupling:

$$\frac{C_f}{2\Lambda}(\partial_\mu\phi)\bar{f}\gamma^\mu\gamma^5 f = -\frac{C_f m_f}{\Lambda}i\bar{f}\gamma^\mu\gamma^5 f + \cdots \tag{11}$$

where the dots indicate terms proportional to $F\tilde{F}$. The decay widths to charged fermions are given by

$$\Gamma(a \to f\bar{f}) = \frac{m_f^2 m_a |C_f^2|}{4\pi\Lambda^2}\sqrt{1 - \frac{4m_f^2}{m_a^2}}. \tag{12}$$

Analogous to scalars, the effective ALP coupling to neutrons can be written as

$$\mathcal{L} \in L_{kin} + \lambda_{eff}n\chi\gamma^5\phi. \tag{13}$$

A comprehensive account of UV-complete models and their phenomenological consequences is left for future work. In principle, since the bosons in our case are heavy, the most-general Lagrangian will contain interaction terms involving not only photons and leptons, but hadrons as well.

The decay widths for pseudoscalars to diphotons are given by

$$\Gamma(a \to \gamma\gamma) = \frac{|C_\gamma^2|}{4\pi\Lambda^2}m_a^3. \tag{14}$$

The lifetime is

$$\tau(a \to \gamma\gamma) = 1/\Gamma(a \to \gamma\gamma \times f) \tag{15}$$

where $f$ is the conversion factor from $\text{GeV}^{-1}$ to seconds. From the estimates derived above, for a boson mass of 1 MeV, a lifetime of $\tau \geq \simeq 10^{11}$ years, and if this is only decay-channel-relevant, the effective coupling, $g_{eff} = C_\gamma/\Lambda \leq 10^{-17}$. Note that a lifetime of $10^{11}$ years is about $10^{18}$ seconds. The lifetime of the universe is about $10^{18}$ seconds, and therefore, this boson is cosmologically stable and should add to the total relic density of the universe. The exact amount of dark matter density depends on the co-efficient $C_f$, as well as the decay constant $\Lambda$. Typically in axion-like models, like the ones considered here, we can obtain a significant fraction of the dark matter with a $\mathcal{O}(1)$ misalignment angle.

There are, however, significant constraints of models of this class. For scalars and pseudoscalars, one of the strongest constraints at this mass originates from the consideration that photons produced during ALP decays when the universe is transparent should not exceed the total extragalactic background light (EBL) [104]. For pseudoscalar ALPs, this

limits the lifetimes to $\tau \geq 10^{23}$ seconds, such that the effective ALP coupling is restricted $g_{eff} \leq 10^{-19}$. Furthermore, X-rays produced from ALP decays in galaxies must not exceed the known backgrounds. This limits $\tau \geq 10^{25}$ seconds, leading to an effective coupling $g_{eff} \leq 10^{-20}$ [104].

*4.2. Spin-1*

While the decay to a photon has been ruled out, a possible solution is that the spin-1 boson can be a dark (kinetically) mixed photon. The massless part of the most-general theory of two $U(1)_{a,b}$ Abelian gauge bosons can be written as:

$$\mathcal{L} = -\frac{1}{4} F_{a\mu\nu} F_a^{\mu\nu} - \frac{1}{4} F_{b\mu\nu} F_b^{\mu\nu} - \frac{\epsilon}{2} F_{a\mu\nu} F_b^{\mu\nu} \tag{16}$$

The masses of these can be obtained via a Stueckelberg mechanism or via a spontaneously broken gauge symmetry:

$$\mathcal{L}_m = \frac{1}{2} M_a^2 A_\mu^a A^{a\mu} + \frac{1}{2} M_b^2 A_\mu^b A^{b\mu} + M_a M_b A_\mu^a A_b^\mu \tag{17}$$

Consider a hypercharge mixing with the usual photon:

$$\mathcal{L} = \frac{\epsilon}{2 \cos \theta_W} \tilde{F}'_{\mu\nu} B^{\mu\nu} . \tag{18}$$

Then, the effective Lagrangian becomes

$$\mathcal{L} \in e\epsilon J_\mu A'_\mu + e'\epsilon \tan \theta_W J'_\mu Z_\mu + e' J'_\mu A'_\mu , \tag{19}$$

where $J'_\mu$ and $e'$ are the dark sector current and the dark photon coupling to the dark sector. Once the Z boson is integrated out, we can see that the coupling of the dark photon to SM fermions is proportional to $e\epsilon$, i.e., millicharged dark photons, which are constrained from various sources. The effective coupling to neutrons can be written as

$$\mathcal{L} \in e\epsilon(n\sigma^{\mu\nu}\chi F'_{\mu\nu}) . \tag{20}$$

Below the two-electron threshold, the constraints on dark photons originate from stellar cooling bounds and from the Xenon-1T experiments [105,106]. The constraints on the kinetic mixing parameter is $\epsilon \leq 10^{-13}$ for $m_{A'} \simeq 1 MeV$.

If the dark photon is extremely light, if produced non-thermally, like a condensate, it is like the axion with a misalignment mechanism. In this case, the mass is generated by the Stueckelberg mechanism, and the generation of the relic follows like the usual axion.

Additionally, in most relevant models, the dark photon is accompanied by dark fermions. Here, the dark photon can account for the relic density through a freeze-in mechanism within the dark sector or through feeble couplings to SM particles.

## 5. Conclusions

In this work, we explored the consequences of neutron decays into a dark sector inside neutron stars. Working on the hypothesis that $n \rightarrow \chi + \phi$, we analysed the feasibility of this decay by studying the effect of the corresponding equation of state on the properties of the neutron star, including its mass, radius, and tidal deformability. We then focused on the possibility that $\phi$ remains trapped inside the star, leading to heating. We concluded that, if $\phi$ has a mass of around 1 MeV, based on the observations of the luminosity of stars as a function of age, $\phi$ must have a lifetime greater than $10^{11}$ years. Finally, we studied the consequences of $\phi$ coupled to standard model particles within simplified ALP-like models. An expanded work with cosmological consequences, as well as a study of UV completions are left for future work.

**Author Contributions:** All authors have contributed equally to the work. All authors have read and agreed to the published version of the manuscript.

**Funding:** This study has been supported by a University of Adelaide International Scholarship (WH) and by the Australian Research Council through the ARC Centre for Dark Matter Particle Physics (CE200100008).

**Conflicts of Interest:** There are no conflicts of interest.

## Note

1    Note that recent observations of pulsars PSR J0030+0451 [45] and PSR J0740+6620 [46] constrain their masses and radii to be 1.34 $M_\odot$ and 12.71 km and 2.072 $M_\odot$ and 12.39 km, respectively. See also [47,48].

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
