# Peer review of "Constraining Dark Boson Decay Using Neutron Stars"

_universe, doi:10.3390/universe9070307_

Round 1
Reviewer 1 Report
exceptionally low quality of the manuscript. it can't be improved and should be rejected.
Author Response
I am disappointed to receive this response. However the other referees and the editor have encouraged the draft, so we will respond to those comments.
Reviewer 2 Report
A very interesting paper, linking the uncertainties in neutron lifetime to existence of invisible modes of neutron decay and its possible astrophysical consequences, in the structure and evolution of neutron stars, in particular. The paper is well written and if authors make proper spell check and remove all the grammatic misprints and errors, can be recommended for publication. Some comments on their hypothetical long-living particle as a component of dark matter would be useful to add. After such minor revision the paper can be recommended for publication.
Author Response
We thank the referee for the encouraging remarks. Please note the following responses,
- We have run a spell check and sorted out the grammatical errors as best as we can.
- We have added a couple of sentences on the dark sector particles as a component of the total DM relic density in section 4.1 and section 4.2
Reviewer 3 Report
In this work the authors explore the consequences of neutron decays into a dark sector inside 240 neutron stars. Working on the hypothesis that n → χ + φ, they analyze the feasibility of this decay by studying the effect of the corresponding equation of state on the properties of the neutron star, including its mass, radius and tidal deformability. They consider the possibility that the φ remains trapped inside the star leading to heating and claim that if the φ has a mass of around 1 MeV, based upon observations of luminosity of stars and as a function of age, the φ must have a lifetime greater than 1011 years.
The authors argue that interacting dark matter increases the mass limit on neutron stars to 2 solar masses
as required by experiment. They also study the possible effective lagrangians for dark particles .
there are several misprints, for example no definition of sigma (meson?) field before eq. 3 where it is used,
or Fig. (??) on page5. Altogether this is an interesting paper that can be published in the present form.
Author Response
We thank the referee for the encouraging comments. We note the criticism of the referee about the Sigma meson. However, this is pretty standard literature and the relevant details can be found in the references cited.